# Active Case Finding for Tuberculosis in India: A Syntheses of Activities and Outcomes Reported by the National Tuberculosis Elimination Programme

**DOI:** 10.3390/tropicalmed6040206

**Published:** 2021-11-30

**Authors:** Sharath Burugina Nagaraja, Pruthu Thekkur, Srinath Satyanarayana, Prathap Tharyan, Karuna D. Sagili, Jamhoih Tonsing, Raghuram Rao, Kuldeep Singh Sachdeva

**Affiliations:** 1Department of Community Medicine, Post Graduate Institute of Medical Sciences and Research, Employees State Insurance Corporation Medical College, Bengaluru 560010, India; 2The Union, South East Asia Office, New Delhi 110016, India; ssrinath@theunion.org (S.S.); ksagili@theunion.org (K.D.S.); jamhoih.tonsing@theglobalfund.org (J.T.); Kuldeep.Sachdeva@theunion.org (K.S.S.); 3BV Moses Centre for Evidence-Informed Health Care, Clinical Epidemiological Unit, Christian Medical College, Vellore 632002, India; prathaptharyan@gmail.com; 4The Global Fund, 1218 Geneva, Switzerland; 5Central TB Division, Ministry of Health and Family Welfare, New Delhi 110001, India; raor@rntcp.org

**Keywords:** tuberculosis, active case finding, diagnostic algorithm, number needed to screen

## Abstract

India launched a national community-based active TB case finding (ACF) campaign in 2017 as part of the strategic plan of the National Tuberculosis Elimination Programme (NTEP). This review evaluated the outcomes for the components of the ACF campaign against the NTEP’s minimum indicators and elicited the challenges faced in implementation. We supplemented data from completed pretested data proformas returned by ACF programme managers from nine states and two union territories (for 2017–2019) and five implementing partner agencies (2013–2020), with summary national data on the state-wise ACF outcomes for 2018–2020 published in annual reports by the NTEP. The data revealed variations in the strategies used to map and screen vulnerable populations and the diagnostic algorithms used across the states and union territories. National data were unavailable to assess whether the NTEP indicators for the minimum proportions identified with presumptive TB among those screened (5%), those with presumptive TB undergoing diagnostic tests (>95%), the minimum sputum smear positivity rate (2% to 3%), those with negative sputum smears tested with chest X-rays or CBNAAT (>95%) and those diagnosed through ACF initiated on anti-TB treatment (>95%) were fulfilled. Only 30% (10/33) of the states in 2018, 23% (7/31) in 2019 and 21% (7/34) in 2020 met the NTEP expectation that 5% of those tested through ACF would be diagnosed with TB (all forms). The number needed to screen to diagnose one person with TB (NNS) was not included among the NTEP’s programme indicators. This rough indicator of the efficiency of ACF varied considerably across the states and union territories. The median NNS in 2018 was 2080 (interquartile range or IQR 517–4068). In 2019, the NNS was 2468 (IQR 1050–7924), and in 2020, the NNS was 906 (IQR 108–6550). The data consistently revealed that the states that tested a greater proportion of those screened during ACF and used chest X-rays or CBNAAT (or both) to diagnose TB had a higher diagnostic yield with a lower NNS. Many implementation challenges, related to health systems, healthcare provision and difficulties experienced by patients, were elicited. We suggest a series of strategic interventions addressing the implementation challenges and the six gaps identified in ACF outcomes and the expected indicators that could potentially improve the efficacy and effectiveness of community-based ACF in India.

## 1. Introduction

Tuberculosis (TB) remains a major public health problem of global concern. In 2019, of the 10 million people estimated to have developed TB worldwide, only 7.2 million people were notified by the National TB Programs (NTP), indicating that around 2.8 million people with TB (28% of the estimated disease burden) were undetected or were detected but not notified [1]. Detecting TB through passive case finding (PCF) results in diagnostic delays and the suboptimal detection of TB patients in low- and middle-income countries with a high TB burden due to geographic and/or socioeconomic barriers in accessing health facilities [2].

In its ‘End TB Strategy’, the World Health Organization (WHO) advocated for the ‘systematic screening’ of high-risk population subgroups to increase TB case detection [3]. ‘Systematic screening’ includes provider-initiated systematic screening for symptoms and signs of TB disease at health facilities or outside health facilities or both [4]. Systematic screening for TB disease at health facilities is also called ‘intensified case finding’. Systematic screening for TB disease outside health facilities is called ‘enhanced case finding’ or ‘active case finding’, depending on whether the engagement with the target high-risk population occurs at the group or individual level. Educating high-risk groups about TB disease and advising those with symptoms to visit health facilities for diagnosis and treatment is called ‘enhanced case finding’ (ECF). Proactively screening all individuals within high-risk groups outside health facilities for TB symptoms and linking those with symptoms to TB diagnostic services with an intention to diagnose and treat TB cases is termed ‘active case finding’ (ACF).

Many systematic reviews have concluded that systematic screening leads to increased TB case detection compared to PCF [5,6,7,8,9,10]. The beneficial effects largely pertain to intensified or targeted case finding among people at high risk for TB in whom the yield of TB cases was high [6,7]. Community-based ACF (or combined ACF and ECF) activities in the general population had uncertain individual and community-level effects, and the benefits of an earlier diagnosis by community-based ACF activities on patient outcomes and transmission were unclear [8]. However, more recent observational studies and controlled trials in settings with a high TB prevalence indicate that community-based ACF activities could reduce the diagnostic delay, limit out-of-pocket expenditure and reduce TB incidence [11,12,13].

Since the benefits of community-based ACF depend on a number of factors, any decisions on ACF components should be based on evidence of an acceptable yield of microbiologically confirmed TB, using diagnostic algorithms to increase the case finding efficiency by considering the expected prevalence, estimated diagnostic test accuracy and the resource availability for specific settings [6,10,14].

### Active Case Finding (ACF) Campaign under the National Tuberculosis Elimination Programme

ACF activities were initiated in India in 2009 by nongovernmental organisations (NGOs). ACF activities have expanded since 2013. In 2017, the Revised National TB Control Programme (RNTCP) for India emphasized in its “National Strategic Plan for Tuberculosis Elimination—2017–2025”, a series of ACF activities to be implemented in a campaign mode to complement PCF strategies. The key component of ACF in the National Tuberculosis Elimination Programme (NTEP, then renamed RNTCP) involves a nationwide community-based ACF campaign that mobilizes almost the entire general health system to conduct house-to-house screening for TB symptoms in mapped vulnerable target populations for two weeks thrice in a year [15]. Other components include using expanded definitions of TB symptoms, chest X-rays as a screening test and rapid molecular tests upfront [15]. A guidance document, updated in 2017, provides detailed information for implementing ACF activities, as well as the targets and quality indicators to monitor the success of the programme [16].

Table 1 details the expected indicators set by the NTEP for the programme managers of the ACF campaign [16].

Figure 1 depicts the flow of activities envisaged in the ACF campaign in the NTEP guidance document. Each set of activities provides potential intervention points that could be utilised strategically to facilitate TB case detection.

Considering the effort, money and resources involved in implementing community-based ACF activities, we undertook a systematic review to provide timely information about the potential benefits of ACF in India. We sought this information from two sources: (a) information from state NTEP programme managers and implementing partner organizations and (b) data from published and unpublished studies that provided details of the cascade of screened populations as part of community-based ACF campaigns. The results from the latter source, largely from non-programme ACF activities conducted in India, will be reported in a separate publication. The review protocol was registered in PROSPERO, the international prospective register of systematic reviews, on 26 August 2020 and assigned the registration number: CRD42020199854 [17].

## 2. Materials and Methods

We followed the guidance provided in the PRISMA statement [18] in developing the protocol and reporting this synthesis of ACF activities and outcomes reported by the NTEP.

### 2.1. Types of Studies

We sought programme data from 2017 to 2020 of ACF activities in India from state NTEP managers and data from 2013 to 2020 from implementing partner agencies about ACF activities supported through the NTEP. The Central TB Division, Ministry of Health, Government of India facilitated this process.

### 2.2. Participants and ACF Activities

We sent a structured, pretested data proforma (Appendix A) to the programme managers. We requested details of community-based ACF activities targeting the general population, urban slums, urban non-slums, rural areas, hard-to-reach areas, tribal populations, migrant populations, drug users, household TB contacts and paediatric and elderly populations. Data from ACF programmes solely dealing with TB case finding in institutional groups, such as prisoners, healthcare workers, occupational risk groups and patient groups such as people with diabetes seeking care in hospitals or clinics, were not sought.

### 2.3. Types of Outcome Measures

The data proforma sought:A description of the ACF programmes and the diagnostic algorithms used to detect TB cases;The outcomes of ACF activities, including the number (and proportion) of people: (a) screened from the vulnerable target population mapped; (b) identified with presumptive TB; (c) tested for TB at the district medical/microscopy centres; (d) diagnosed with all forms of TB (positivity rate or yield) through sputum microscopy, chest X-ray and GeneXpert (cartridge-based nucleic acid amplification tests or CBNAAT); (e) initiated on anti-TB treatment (treatment initiation rate) and (f) completing treatment (treatment completion rate, loss to follow up and mortality). From this data, we estimated (g) the number needed to screen (NNS), which is the number of individuals who were screened to identify one person diagnosed with TB. We also sought (h) data on the impact of ACF on TB notification.The challenges encountered during the process of community-based ACF.

### 2.4. Search Methods

Two authors (KS and SBN) obtained the contact details of all the organisations and agencies that were supported by the RNTCP/NTEP to implement community-based ACF from January 2013 to December 2020 from the Central TB Division (CTD). We mailed the data proformas in the first week of November 2020 to the programme managers and partner agency contacts. We sent reminders for missing data forms and closed data collection by 30 December 2020. We did not receive data for ACF activities from many NTEP programme managers, since they were involved in the response to the SARS-CoV-2/COVID-19 pandemic. We supplemented the information provided by the state NTEP programme managers with national summary ACF data for all states and union territories from the annual TB reports of the NTEP [19,20,21,22].

### 2.5. Data Management and Analysiss

One of us (SBN) extracted and collated data from the returned data proformas. PT and SS independently checked the extracted data, and all authors reviewed and discussed the data. We evaluated the adequacy of reporting in accordance with the TIDieR-PHP reporting guidelines for population health and policy interventions [23].

Since the data were heterogeneous, we tabulated our results in accordance with the synthesis without meta-analysis (SWiM) guidelines [24]. We assessed the proportions mapped, screened, identified, tested, diagnosed and treated against the expected proportions set by the NTEP for the ACF programme (Table 1). We derived the NNS (from the numbers screened/number diagnosed with TB) for each year for each state and partner agency [25]. We used the NTEP indicators as a framework to assess the possible associations between the proportions completing relevant parts of the ACF cascade, TB detection rates and the NNS.

We used the information regarding implementation challenges elicited from the programme managers in the returned data proformas and additional information gathered from discussions with the programme managers and partner agencies. We listed them under the broad themes of challenges in implementing ACF activities related to the health system, healthcare provision and those experienced by patients. For further details about the methods used in data management and analyses, see Appendix A.

## 3. Results

### 3.1. Respondents

Programme managers from nine states, two union territories and five partner agencies returned completed proformas from among the 28 states, eight union territories and eight partner agencies approached for ACF data (Figure 2).

The five NTEP implementing partner agencies were: International Union Against Tuberculosis and Lung Disease (The Union): Project Axshya, the Indian Council for Medical Research (ICMR): Project TIE-TB, the Karnataka Health Promotion Trust (KHPT): Tuberculosis Health Action Learning (THALI) Project, World Health Partners: THALI Project and World Vision India: Project Axshya. Three other partner agencies approached reported no community-based ACF activities in the stipulated time frame.

Though many states did not return completed data proformas, the partner agencies that responded conducted ACF in many unrepresented states.

### 3.2. Overall Data Quality and Completeness

Four states (Bihar, Gujarat, Karnataka and Maharashtra) provided us with Excel data sheets (in addition to the data proformas) with district and state-wide results of the various rounds of ACF activities conducted from 2017 to 2019. This permitted a comprehensive assessment of the reporting of their ACF activities. The two union territories provided consolidated data for each round of ACF for 2018 and 2019. The remaining states also provided consolidated data (yearly data for 2017–2019—two states and combined data for 2018 and 2019—one state). Two states provided data only for 2017.

The ICMR provided an interim progress report with district-wide data from the five states involved in a 11-month intervention in tribal populations (TIE-TB). The other implementing partners provided combined data from all the years of their respective projects.

We used the available data provided by the states for three years (2017–2019), since the data for 2020 were incomplete or not available.

For 2020, we used the summary data reported in the India TB annual report [22]. Since the ACF activities during 2020 may have been disrupted due to the pandemic, we compared the patterns in ACF activities for two pre-pandemic years from the India TB reports for 2018 [20] and 2019 [21]. Detailed summary ACF data for 2017 were not available in the corresponding annual report [19].

Deficiencies in reporting were present in about 50% of the returned data proformas (Appendix A). These were primarily for numerical data for the vulnerable target populations mapped; the proportions screened who were identified with presumptive TB; the proportions with presumptive TB who were tested for TB and the proportions that underwent sputum tests, chest X-rays or CBNAAT. None of the respondents provided numerical data for sputum smear-negative cases or for false positives.

However, all respondents provided numerical data for those with positive tests (sputum, chest X-ray and CBNAAT) and the numbers diagnosed with TB (all forms). It was, therefore, possible to calculate the diagnostic yield (TB prevalence). The NNS could also be estimated for ACF activities in all the states and union territories.

The numbers initiating treatment were unavailable from the data provided by five states. Only three states and the ICMR project provided data on treatment outcomes. The impact of ACF on TB notification was not provided by the states. Among the implementing partners, this was only available for the ICMR TIE-TB project.

### 3.3. Details of ACF Activities

#### 3.3.1. Frequency and Duration of ACF Activities

The NTEP advocates three rounds of ACF lasting 15 days each year (45 days per year). States used varying strategies for ACF, from biannual rounds for 15 days (or longer), monthly rounds for a variable number of days, ACF activities throughout the year or for one day in a year. The total duration in the states was fewer than the 45 days per year suggested by the ACF campaign. The recommended duration was exceeded by the implementing partners, since ACF occurred daily for the duration of their projects.

#### 3.3.2. Mapping Vulnerable Populations and Selecting Target Areas

Mapping areas for the ACF activities and selecting the target vulnerable populations to screen was based on the guidance provided by the NTEP [16] and influenced by local knowledge about geographical and occupational vulnerabilities. The prevalence of TB in the populations mapped did not appear to be a formal part of this exercise, except for one partner agency that estimated a variable prevalence of 200–300 per 100,000 population in their areas of activity. The implementing partners worked exclusively among vulnerable high-risk groups, while ACF in the states covered a wider range of vulnerable populations.

#### 3.3.3. Types of ACF Activities

All ACF activities reportedly followed the NTEP guidance and provided health education prior to interviews of household members. Field staff were not always able to screen all members of the household. Many interviewed only the available members, or the head of the household, about symptoms in others. The frequency and proportion of revisits to screen members missed during the initial visits were unclear. In the ICMR’s TIE-TB project, the district TB officers prepared a monthly schedule that was shared with the local primary health centre (PHC) staff to mobilise people who had been screened and presumed to have TB about the monthly visits of the mobile diagnostic vans.

#### 3.3.4. Personnel Engaged for ACF

Apart from community health workers, multipurpose health workers from the general health system and from rural childcare centres (anganwadi workers), workers from the accredited social health activists (ASHA) programme, school wardens, teachers and NGO volunteers were engaged for ACF under the NTEP. Training for ACF personnel using standard guidelines and algorithms was provided over one day by the district programme managers and supervised by the district TB officer and the medical officer for TB control.

The implementing partners recruited and trained project field staff exclusively for ACF activity. Each partner employed different models in delivering ACF activities. For example, The Union’s Project Axshya worked through sub-recipient partners who engaged community volunteers to conduct ACF, with periodic reviews by district and state NTEP managers and external evaluation by the funding agencies. KHPT’s THALI project used community health workers and community structures to implement ACF and trained 19,272 accredited social health activists (ASHA) in ACF activities

#### 3.3.5. Incentives for ACF Personnel and Support for Activities

In larger states, volunteer ACF personnel were provided incentives ranging from 100 to 500 Indian rupees (INR) per day of ACF activity (around 1.3–6.5 US$); one state also provided the team INR 500 per confirmed case. In some states with difficult terrains, the NTEP staff were also provided incentives. One of the partner agencies paid their project staff INR 10 per household screened, one provided incentives only after treatment initiation was documented and three did not provide incentives.

ACF personnel in most states and partner agencies collected sputum samples and transported them to the diagnostic facilities. Under The Union’s Project Axshya, this activity was incentivized with INR 100 per sample transported; however, to ensure the sputum quality, there were limits on the number of samples transported and the sputum positivity rate. In most states, ACF personnel also provided referral forms, travel support or vouchers to people to get chest X-rays done.

The ICMR TIE-TB project exclusively used mobile TB diagnostic vans equipped with sputum microscopy and digital X-ray facilities that visited remote tribal villages once a month. Most states reported that mobile vans for X-rays and for other ACF activities were available, and in one state, the mobile van also had facilities to perform CBNAAT. It was unclear from the data provided how frequently these vans were available and were utilised in ACF activities.

#### 3.3.6. Diagnostic Algorithms Used

Following the initiation of the ACF campaign in the NTEP in 2017, Xpert MTB/Rif tests were deployed up to the subdistrict level in all the states by 2018. By 2019, all states, barring some districts in some small states, reported using the NETP advocated algorithm for sputum smear microscopy and chest X-ray followed by CBNAAT and the TB case definitions provided in Figure 1. Some also used variations of the diagnostic algorithms, such as sputum microscopy combined with chest X-ray or X-ray and CBNAAT in parallel, while one state performed only smear microscopy and CBNAAT.

The breakdown of the proportion of TB cases diagnosed using each algorithm was not uniformly available from the data provided by the states.

### 3.4. ACF Activities and Outcomes from Responding State and Union Territories

Table 2 details the activities and outcomes of the active case finding (ACF) campaigns conducted in the states and union territories in India from the available data provided by the NTEP managers for the years 2017–2019.

### 3.5. ACF Activities Provided by Partner Agencies

Table 3 provides details of the activities and outcomes provided by the agencies that implemented ACF in partnership with the NTEP.

### 3.6. ACF Outcomes for All States and Union Territories in India for 2020

Table 4 reproduces summary data on ACF activities provided to the NTEP by all the states and union territories in India for the year 2020 that were published in the annual report of the NTEP [22].

The state-wise breakup of the proportions diagnosed with TB by different diagnostic tests was not reported.

### 3.7. ACF Outcomes for States and Union Terrritories Compared to the Expected Indicators for ACF Set by the NTEP

We used the numerical data provided in these tables (Table 2, Table 3 and Table 4) and national ACF data for 2018 and 2019 (Appendix A) to evaluate the screening, diagnostic and treatment activities undertaken by the states and union territories and their implementing partners against the expected indicators envisioned by the NTEP for ACF (Table 1).

#### 3.7.1. Vulnerable Target Population Mapped and Screened

The NTEP expects that 110,000 per million vulnerable population (11%) should be mapped for community-based screening.

The national ACF data for 2020 (Table 4) revealed that, despite the disruption caused by the ongoing SARS-CoV-2/COVID-19 pandemic, nearly 25% of India’s 13,378 million population was mapped for screening. This proportion ranged across the states from <1% to 100% (median 10.7%). In 2020, 53% of the states and union territories (Table 4) and 61% in 2019 (Appendix A) met or exceeded the NTEPs expected indicators for mapping vulnerable populations.

The NTEP expects that >90% of the mapped target vulnerable population should be screened for symptoms of TB.

In 2020, around 51% (range <1–100%; median 37%) of those mapped across the states were screened (Table 4). Eleven states and union territories screened more than 75% of the target population in 2020 but only in 4/34 (12%) with the available data did this exceed the 90% expectation of the NTEP. For 2018 (Appendix A) and 2019 (Appendix A), the proportion of states and union territories meeting or exceeding this NTEP indicator was 19% and 16%, respectively.

#### 3.7.2. Proportions Undergoing Diagnostic Tests for TB among Those Screened and in Those with Presumptive TB

The NTEP expects that around 5% of people in the community with TB symptoms will be identified through house-to-house screening.

However, the proportions identified with presumptive TB through symptom screening in the states and union territories is not reported in the annual TB reports (Table 4, Appendix A).

From the available data for 2017–2019 from the state programme managers, the proportion identified as having presumptive TB from among those screened ranged from <1% in three states, 5% to 10% in three states and 20% to 67% in two others (Table 2). These proportions varied across the three years of reporting. The proportion of responding states that met or exceeded the NTEP expectation for presumptive TB cases identified through ACF ranged from 14% in 2017, 83% in 2018 to 0% in 2019 (Table 2).

The NTEP has not set minimum indicators for the proportion to be tested for TB among those screened (Table 1).

However, the data are reported annually for the proportions tested from among those screened. From 2018 to 2020, <1% of those screened for symptoms of TB in ACF activities across the states and union territories in India underwent TB diagnostic tests (Appendix A and Table 4).

The NTEP expects that >95% of those identified with presumptive TB will be tested for TB.

This information is not reported in the annual TB reports (Appendix A and Table 4). The available data from the responding states and union territories revealed that 40% (two out of five) in 2017, 29% (two out of seven) in 201 and 17% (one out of six) in 2019 met or exceeded the NTEP indicators for the proportions with presumptive TB who were tested for TB (Table 2).

#### 3.7.3. Diagnostic Algorithms Used and the Proportions Tested with Sputum Smear Microscopy, Chest X-ray and Xpert MTB/Rif (CBNAAT)

The NTEP expects that 5% (minimum >2% to 3%) of those tested would have sputum smear-positive test results.

Sputum smear positivity rates were not available from the national summary tables for the ACF programme (Appendix A and Table 4).

Sputum smear positivity rates exceeded the 2% minimum expected in 8/10 responding states and union territories for all or most of the years reported (Table 2). The higher smear positivity seen in the partner agencies data (5.3–16.5%) reflect their focus on screening populations at a higher risk of undiagnosed TB through ACF activities that occurred throughout the year (Table 3).

The NTEP expects that >90% of sputum smear-negative TB patients will be examined by chest X-ray or CBNAAT or both.

This data is not provided in the annual TB reports (Appendix A and Table 4).

Most responding states did not report the number of people with sputum smear-negative results who underwent further testing with chest X-rays or CBNAAT or both (Table 2 and Table 3). Where this could be estimated, the data revealed that only 29% (two out of seven) of the responding states in 2017, 50% (three out of six) in 2018 and 20% (one out of five) in 2019 performed chest X-ray or CBNAAT on >90% those with sputum smear-negative results (Table 3).

#### 3.7.4. Proportions Diagnosed with All Forms of TB (Diagnostic Yield)

The NTEP expects that at least 5% of people undergoing diagnostic tests in ACF programmes would be diagnosed with TB (all forms).

Based on the available data from the states and union territories (Table 2), this expectation was met or exceeded by 57% (four out of seven) in 2017, 33% (two out of six) in 2018 and 60% (three out of five) in 2019. The implementing partners exceeded this target for all years of their activities (5.3–48.4%; Table 3). Data from more complete nationwide datasets revealed that this expectation was achieved or exceeded in 30% (10/33) of the states and union territories in 2018 (Appendix A), 23% (7/31) in 2019 (Appendix A) and 21% (7/34) in 2020 (Table 4).

#### 3.7.5. Proportions Initiating and Completing Anti-TB Treatment

The NTEP expects that >95% of TB patients diagnosed through ACF activities should be initiated on anti-TB treatment.

The proportions started on anti-TB through ACF activities in India from 2017 to 2020 were not available from the ACF annual reports published by the NTEP [19,20,21,22].

The NTEP programme managers stated that, although all diagnosed patients were notified about the NTEP via Nikshay (https://nikshay.in, accessed on 21 January 2021), the national web-based TB information management system, they could not retrieve treatment information specific to their ACF patients, as Nikshay does not have a dedicated ACF module. Four states and one union territory provided data from their own records of the numbers initiated on treatment (Table 2), and treatment completion ranged from 89% to 100%. The partner agencies reported that 96–100% were started on anti-TB treatment, but none provided data for the treatment outcomes (Table 3).

#### 3.7.6. The Number Needed to Screen (NNS)

The NTEP performance indicators do not include the NNS.

The NNS varied within the states for each year of ACF activity and between the states in the data from the state programme managers (Table 2), implementation partners (Table 3) and national ACF data for 2018–2020 (Appendix A and Table 4).

The data uniformly demonstrated that the higher the proportions that are tested for TB among those screened, and the more accurate the tests used to diagnose TB are, the lower the NNS. For example, in 2020 (Table 4), the state with the lowest NNS (Jharkhand) tested 71% of those screened and diagnosed TB in 17.6% of them, resulting in an NNS of 8. The highest NNS of 23,356 was seen in Lakshadweep in the same year, when 0.7% of those screened were tested, and TB was diagnosed in only 0.3% of those tested. Additionally key in this relationship is the risk of TB in the proportions screened and tested.

#### 3.7.7. The Impact of ACF on TB Notification

The NTEP performance indicators do not include the assessment of TB notifications due to ACF programmes.

Data for the impact of ACF on TB notifications in the states and union territories were not provided in the annual TB reports [19,20,21,22].

Among the partner agencies, only the ICMR TIE-TB project provided data that assessed the impact of the mobile diagnostic units. Of the 24,043 total TB notifications from all sources from October 2017 to June 2018 from the five states that were covered by the project, 3816 (16%) were notified by the mobile diagnostic vans. This proportion varied across the five states (4–25%; median 18%).

The project also estimated that the mean out-of-pocket expenditure for treatment (travel, consultation, investigations, medicines and ancillary costs such as food) was reduced by 78% for patients serviced by mobile vans (average cost INR 255) compared to if they had availed themselves of services through standard government facilities (average cost INR 1163). The reduction in personal expenditure was even greater when compared to treatment at private facilities (average cost INR 6897; 47% spend more than INR 10,000). The project also demonstrated modest reductions in the time to seek consultations, being diagnosed and starting treatment compared to using standard government facilities or private services.

### 3.8. Challenges Faced by Implementors in Implementing ACF

In Table 5, we list the challenges expressed by programme managers in implementing ACF, obtained from the responses in the data proformas returned by the programme managers of the state NTEP programmes and the partner agencies and through discussions with some of them.

## 4. Discussion

India has the highest TB burden in the world. In 2019, India accounted for the largest proportion of people worldwide diagnosed with TB and drug-resistant TB and the largest proportion of under-reported or undiagnosed TB cases [26]. Implementing a successful community-based ACF programme in a country with a population of over 1.3 billion people is a herculean task. This is the first paper to evaluate the available data from India’s ACF programme against the performance indicators set by the NTEP for ACF and to use the NNS to assess the efficiency of ACF in the states and union territories of the country.

### 4.1. The Gaps between the Expected Indicators and Outcomes in India’s ACF Programme

This review identified six gaps in India’s ACF programme where the data for the outcomes fell short of the expected performance indicators.

Strategic gap 1: The deficiencies in the proportions mapped from the vulnerable target populations and those screened among the mapped populations.

Strategic gap 2: The discrepancy in reporting the proportions tested among those screened for TB (which is not among the NTEP’s performance indicators), instead of the proportions tested among those identified with presumptive TB (which is a performance indicator for which national data were unavailable).

Strategic gap 3: The deficiencies in ensuring that over 95% of the people that identified with presumptive TB underwent diagnostic testing.

Strategic gap 4: The lack of data to evaluate whether >90% with negative sputum test results underwent additional diagnostic tests.

Strategic gap 5: The deficits in achieving the NTEP’s minimum expected diagnostic yield of 5% TB cases diagnosed among those tested in the ACF programme.

Strategic gap 6: The lack of data from national reports for the proportions initiating and completing treatment in the ACF programmes (and the resultant lack of data to assess the impact of ACF).

In addition to these gaps, the data in this review demonstrate that if a larger proportion of those screened for TB are tested with accurate diagnostic tests, then the NNS would be lower than it currently is in many state ACF campaigns.

### 4.2. Implications for Potential Interventions to Improve ACF Outcomes and Efficiency

These gaps identify strategic points where various interventions could improve the effectiveness of ACF campaigns.

#### 4.2.1. Improving the Mapping of Vulnerable Populations and Increasing the Uptake of Screening

The gaps identified in mapping vulnerable target populations at a high risk of TB (Strategic gap 1) indicate the need for more accurate and updated TB prevalence data than what is currently available from national and subnational surveys and prevalence studies [27,28,29,30,31]. The challenges experienced by programme managers in mapping vulnerable populations (Table 5) also indicate the need for updated census data and better delineation of the geographical boundaries to be mapped.

One reason identified by programme managers, and echoed in other enquiries [32], contributing to the low uptake of screening in some areas is the perception in segments of the public about TB. ACF programmes that occur only periodically will have less opportunities to influence public opinion. They also will identify fewer people with undetected TB.

If ACF activities in India are to scale up from campaign mode, more sustained ACF activities must be considered. One option is to integrate ACF with other surveillance activities [26]. This was successful in 2000 with the active case search and TB-COVID bidirectional screening that enabled TB notifications in India to increase after the lockdowns were lifted [22]. Scaling up ACF in India also provides an opportunity to align these activities within the broader perspective of the WHO’s multisectoral accountability framework (MAF-TB) [33]. This would also address the risk factors and determinants of TB and enable collaboration with agencies and stakeholders working on the other sustainable development goals [34].

Utilising innovative ways to mobilise and use community networks could also be considered. One such approach is the ‘seed-and-recruit’ approach that has been well-received, was deemed feasible and identified more bacteriologically confirmed cases than one-off ACF activities in the countries that have used this approach [35,36,37].

#### 4.2.2. Better Use of Data Management Systems

Integrating the available information in Nikshay, the national TB patient management system that also serves as the national TB surveillance system, with geographic information system (GIS) mapping, could provide better estimates of TB prevalence (Strategic gap 1). This data would better inform the state NTEP managers while planning ACF mapping and screening activities, especially in urban areas [26,38].

A dedicated ACF module is currently unavailable in Nikshay, and this was perceived as an implementation challenge (Table 5). A module in Nikshay to document all ACF activities, including the actual number of people in households that were interviewed for symptoms and how many individuals were not, would further address strategic gap 1. The Nikshay mobile app could be used to enter this data by authorised NTEP field staff, as is done by community health workers in other high-TB burdened countries with successful ACF programmes [38]. Using the mobile app could also streamline the data captured about ACF activities that currently relies on paper forms in many places.

If ACF activities are linked in Nikshay to the diagnostic investigations performed for each person identified through ACF, it would then be possible to generate data on the sputum samples tested. Sputum smear-positive and smear-negative results could be sent to ACF personnel (even through automated messages) to decide on further tests or to facilitate prompt TB notification and treatment initiation and to assess the risk of false-positive diagnoses resulting from screening (Strategic gaps 3, 4 and 6).

This would also permit ACF personnel and managers to review the proportions that did not undergo further tests, assess the reasons for this and encourage a return for tests, with mobile diagnostic units stationed in convenient locations to facilitate this. This would help to reduce pre-diagnosis dropouts (Strategic gaps 3–5).

Linking details of patients diagnosed with TB by ACF in Nikshay and providing real-time access of this data to ACF personnel would also help reduce post-diagnosis drop-outs and provide data about treatment initiation and completion rates (Strategic gap 6).

This information in the ACF module in Nikshay would also provide granular information at a subdistrict level that could be used to assess the impact of ACF activities on TB notifications and treatment outcomes for patients diagnosed by community-based ACF versus more targeted approaches (Strategic gap 6). This information could also be used to track temporal trends in TB identification from different areas in a district and state that can used in refining mapping and screening activities for future rounds of ACF activities (Strategic gap 1).

The India TB report of ACF activities in 2020 contains a bubble plot of the NNS for each state and each high-risk group that was generated through Nikshay using the data provided by the states [22]. The available NNS data, if linked specifically to ACF activities, could be used to guide strategic planning and implementation decisions to improve the efficiency of the ACF activities.

Making better use of Nikshay for ACF would a cost-effective intervention that will contribute immensely to reducing the six strategic gaps in the ACF cascade identified in this review.

#### 4.2.3. Moving beyond Screening for TB Symptoms

If the aim of ACF is to diagnose people with undetected TB in the community, house-to-house screening for TB symptoms will be insufficient. Many national prevalence surveys across Asia have shown that 40–70% of people detected to have bacteriologically confirmed TB do not report TB symptoms that meet the screening criteria for presumptive TB. Many were detected only because the entire eligible survey population was screened using chest X-rays [39,40,41,42]. In addition, relying on symptom screening would also miss a large proportion of people with subclinical TB who are usually diagnosed by chest X-ray abnormalities or with molecular techniques [43,44].

This implies that careful consideration should be given to expanding the number of people screened who are offered diagnostic tests, irrespective of whether they have symptoms that meet the criteria for presumptive TB. Selecting the asymptomatic people who are offered additional tests needs to be guided by operational research [45]. This will help in addressing strategic gaps 1–3.

#### 4.2.4. Increasing the Diagnostic Yield with ACF

The data in this review does not provide clarity on the diagnostic algorithm that would provide the best yield. The use of mobile diagnostic units with digital X-rays and sputum smear microscopy facilities is a pragmatic alternative with the benefits of getting rapid results, as demonstrated by the TIE-TB project. The WHO recommends the use of computer-aided diagnosis (CAD) for interpreting digital X-rays in screening and triage for TB disease in adults over 15 years of age [26]. The results of operational research should guide the introduction of CAD technologies into scaled-up ACF activities in India.

Expanding the use of Xpert MTB/RIF in ACF programmes is clearly likely to increase the TB notification rates and the numbers initiating treatment [39,46,47]. This expansion is likely to be cost-effective compared to using cheaper tests with lower accuracy [48,49]. This will address strategic gap 5 and also contribute to reducing the NNS.

Screening fewer people but testing more of them with accurate diagnostic tools would increase the diagnostic yield and also reduce the NNS (Strategic gaps 1–5). This strategy should be weighed against the current strategy of setting targets to screen large numbers but testing only a small proportion who meet the symptom criteria [26].

### 4.3. Limitations of the Review

Some of the limitations of this review relate to the data available from the states and union territories and the partner agencies. Not all states and partner agencies provided the data requested. Additionally, there were lacunae in the data proformas returned by the states and partner agencies.

We also made some changes to the review process after the protocol of this review was registered that was necessitated by the data available for evaluation (Appendix A).

The challenges faced in implementing ACF were collated from discussions with the NTEP and partner agency programme managers and the responses provided in the data proformas returned by them. These discussions were limited to the programme managers that we were able to reach and, also, did not necessarily capture the difficulties faced by other ACF personnel. They were also not systematic evaluations using formal qualitative methods. However, they provided valuable insights into some of the strategic gaps identified.

## 5. Conclusions

This review and synthesis of programme activities and outcomes of the ACF campaign launched by the NTEP identified six broad areas where there are gaps between the expectations of the NTEP and the available outcome data from the states and partners implementing ACF. These gaps provided opportunities to intervene strategically, and this review suggests possible interventions that could be considered to improve the efficacy and effectiveness of ACF.

## Figures and Tables

**Figure 1 tropicalmed-06-00206-f001:**
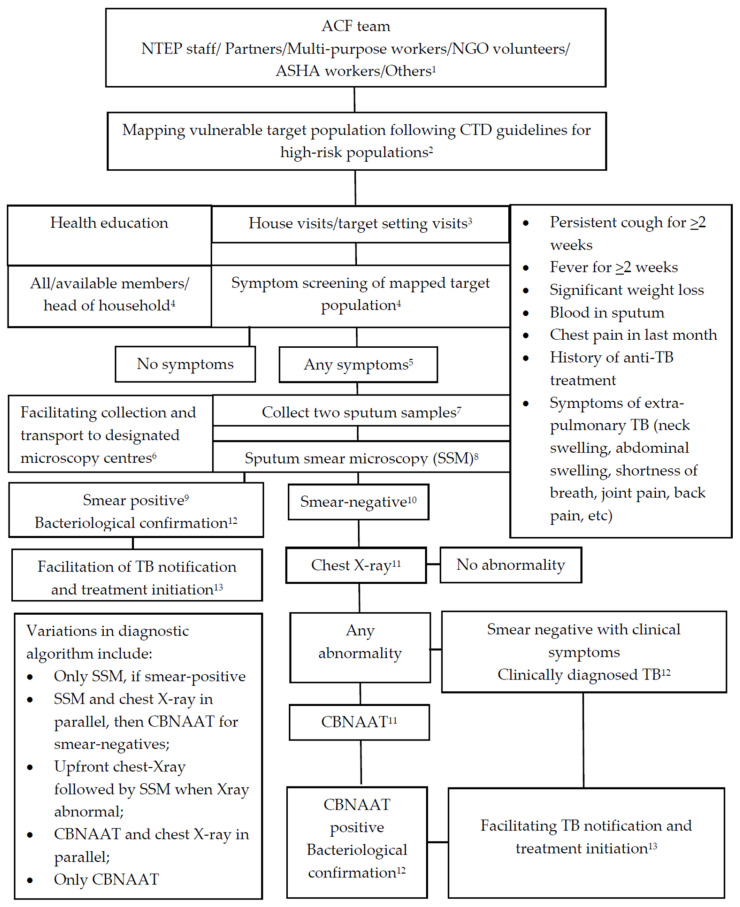
Screening flow chart for active case finding (ACF) in campaign mode under the National Tuberculosis Elimination Programme (NTEP) with intervention points to facilitate case finding and treatment initiation. ASHA: Accredited social health activist; CBNAAT: Cartridge-based nucleic acid amplification test; CTD: Central TB division. **Facilitators**: ^1^ Resources, training, motivation; ^2^ Vulnerable population per million mapped for screening-11%; ^3^ Strategic enumeration, health education, community mobili-zation; ^4^ Setting targets, providing incentives, screening at least 90% of target population; ^5^ At least 5% presumptive TB patients identified through screening; ^6^ Facilitating diagnostic testing, ensuring >95% with presumptive TB get tested; ^7^ Quality control; ^8^ Increased availability, including mobile units; quality control; ^9^ Sputum smear-positives expected: 5% (at least 2–3%); ^10^ Sputum smear-negatives examined by chest X-ray and/or CBNAAT: >90%; ^11^ Increased availability, quality control; ^12^ TB diagnosed (all forms) among those tested: at least 5%; ^13^ Initiated on treatment: >95%.

**Figure 2 tropicalmed-06-00206-f002:**
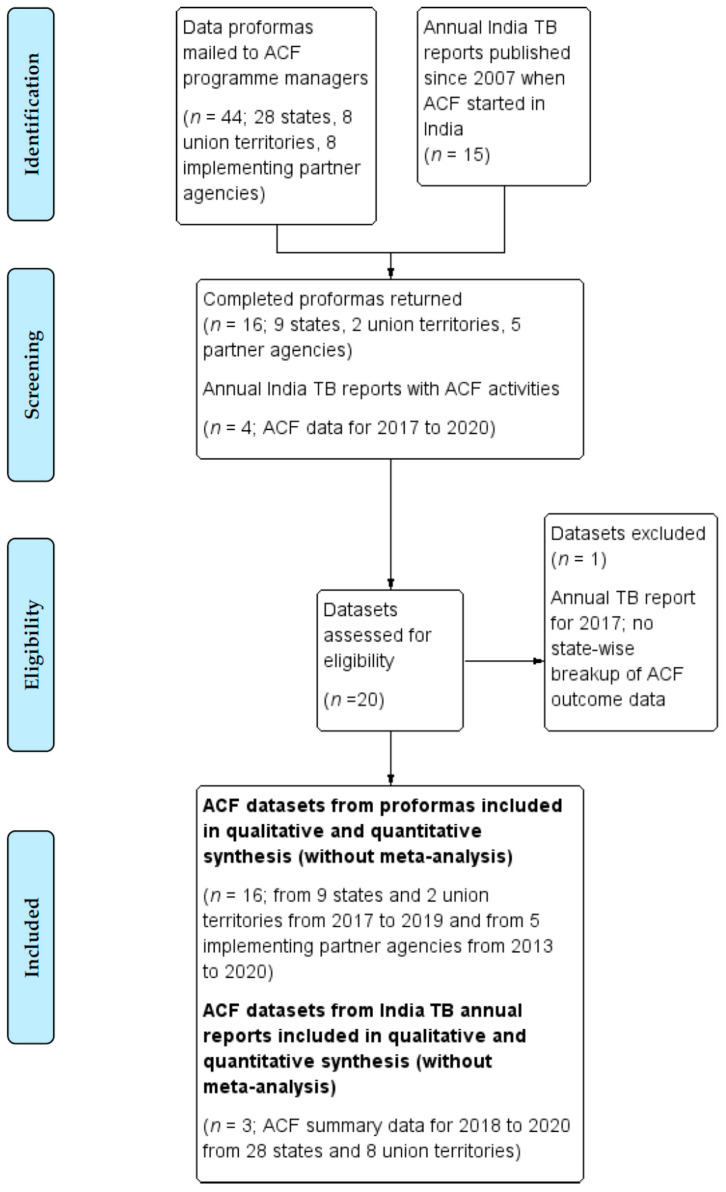
Flow diagram, in accordance with the PRISMA (Preferred Reporting Items for Systematic Reviews and Meta-Analysis) statement, for the identification and selection of data on community-based active case finding (ACF) activities supported by the National Tuberculosis Elimination Programme (NTEP) of India.

**Table 1 tropicalmed-06-00206-t001:** Expected indicators for the active case finding (ACF) campaign in the National Tuberculosis Elimination Programme *.

Indicator	Expected Proportion
Vulnerable population to be mapped per 1 million population	11%
Number in the mapped target population to be screened	>90%
Number with presumptive TB among those screened	5%
Number with presumptive TB patients examined (by smear microscopy, CBNAAT or other investigations)	>95%
Number with sputum smear-positive test results	5% (minimum >2% to 3%)
Number of sputum smear-negative TB patients examined by chest X-ray and/or CBNAAT	>90%
Number with TB diagnosed among those tested	5%
Number of diagnosed TB patients put on treatment	>95%

* Adapted from the Central TB Division: Active TB case finding. Guidance document [16]. CBNAAT = Cartridge based nucleic acid amplification test.

**Table 2 tropicalmed-06-00206-t002:** Activities and outcomes of the active case finding (ACF) campaigns conducted in the states and union territories in India from the available data provided by the National TB Elimination Programme managers (2017–2019).

State/ Union Territory	Year	Target Population Mapped	Numbers Screened (%)	TB Tested in Those with Presumptive TB (%) and among Those Screened [%]	TB Diagnostic Tests			TB Diagnosed (%; 95% CI)	NNS	
					Sputum Positive (%)	X-ray Abnormal (%)	CBNAAT Positive (%)			Anti-TB Treatment Initiated (%)
Andaman & Nicobar	2017	18,526	15,040 (81.1)	11/11 ^a^ (100) [0.7]	11/11 (100)	1/1 (100)	10/11 (90.9)	11 (100; 74.1 to 100)	1367	11 (100)
	2018	1389	46 (3.3)	31/31 ^a^ (100) [67.4]	1/23 (4.4)	0/13 (0)	1/5 (20)	1 (3.2; 0.6 to 16.2)	46	1 (100)
Andhra Pradesh	2018 to 2019	34,220,840	465,223 (1.4)	55,922/55,922 ^a^ (100) [12.0]	4736/55,922 (8.5)	NA	NA	4736 (8.5; 8.2 to 8.7)	98	NA
Bihar	2017	5,650,354	3,033,966 (53.7)	NA	3130/33,754 (9.3)	Nil	Nil	3130 (9.3; 9.0 to 9.6)	969	NA
	2018	2,722,279	1,453,422 (53.4)	NA	816/24,482 (3.3)	Nil	Nil	816 (3.3; 3.1 to 3.6)	1781	NA
	2019	10,298,046	6,141,262 (59.6)	44,858/329,060 (13.6) [0.7]	2583/31,955 (8.1)	921/3974 (23.2)	559/2046 (27.3)	3200 (7.1; 6.9 to 7.4)	1919	NA
Gujarat	2017	14,747,300	4,763,436 (32.3)	37,899/65,059 (58.3) [0.8]	1331/37,899 (3.5)	930/6185 (15.0)	Nil	2261 (6.0; 5.7 to 6.2)	2106	NA
	2018	29,310,663	18,452,680 (63.0)	60,764/79,723 (76.2) [0.3]	1922/60,764 (3.2)	1192/15176 (7.9)	320/4437 (7.2)	3562 (5.9; 5.7 to 6.1)	5180	1856 (52.1)
	2019	59,397,280	37,692,373 (63.5)	77,680/101,304 (76.6) [0.2]	1889/71,039 (2.7)	887/20,269 (4.4)	311/11,892 (2.6)	3087 (4.0; 3.8 to 4.1)	12,210	1931 (62.6)
Karnataka	2017	12,489,357	12,086,328 (96.8)	110,910/110,910^a^ (100) [0.9]	4093/110,910 (3.7)	Nil	Nil	4093 (3.7; 3.6 to 3.8)	2952	NA
	2018	NA	10,265,692 (NA)	90,041/99,946 (90.1) [0.9]	1822/85,408 (2.1)	1914/15,609 (12.3)	372/1715 (21.7)	2957 (2.7; 2.6 to 2.8)	3472	NA
	2019	NA	43,478,614 (NA)	260,157/307,519 (84.6) [0.6]	4205/245,243 (1.7)	4527/42,077 (10.8)	1836/5747 (40.0)	7283 (2.8; 2.4 to 2.7)	5969	NA
Ladakh (Leh & Kargil)	2018	35,798	25,116 (70.2)	462/NA (NA) [1.8]	3/374 (0.8)	0/148 (0)	13/462 (2.8)	13 (2.8; 1.7 to 4.8)	1932	13 (100)
	2019	8996	6199 (68.9)	462/NA (NA); [7.5]	12/205 (5.9)	Nil	1/462 (0.2)	13 (2.8; 1.7 to 4.8)	477	13 (100)
Maharashtra	2017	10,363,469	9,413,295 (90.8)	43,945/55,381 (79.0) [0.5]	1357/43,945 (3.1)	2336/17,663 (15.9)	225/1698 (13.2)	2654 (6.0; 5.8 to 6.3)	3547	2410 (90.8)
	2018	23,479,803	21,281,430 (90.6)	74,634/91,225 (81.5) [0.4]	1925/74,634 (2.6)	4078/25,283 (16.1)	411/5209 (7.9)	3912 (5.2; 5.1 to 5.4)	5440	3845 (98.3)
	2019	95,163,760	87,568,441 (92.0)	192,300/211,850 (90.8) [0.2]	5815/192,300 (3.0)	27,009/145,805 (18.5)	1350/23,570 (5.7)	11,363 (5.9; 5.8 to 6.0)	7707	11,151 (98.1)
Manipur	2017	46,429	31,291 (67.4)	1827/NA (NA) [5.8]	37/1827 (2.0)	0/5 (0)	Nil	37 (2.0; 1.5 to 2.8)	846	NA
Mizoram	2017	16,8028	86,391 (51.4)	2378/NA (NA) [2.8]	14/272 (5.2)	0/5 (0)	47/2106 (2.2)	61 (2.6; 2.0 to 3.9)	1416	61 (100)
Tamil Nadu	2017 to 2019	8,781,657	4,967,754 (56.6)	1,972,878/ 3,343,099 (59.0) [39.7]	NA/1,972,878 (NA)	NA/1,136,568 (NA)	2019 data: 277/5969 (4.6)	6580 (0.3; 0.3 to 0.4)	755	2017 data: 2468/3304 (74.7)
Uttarakhand	2017 to 2019	1,412,700	125,516 (8.9)	10,716/NA (NA) [8.5]	324/10,716 (3.0)	68/432 (15.7)	15/600 (2.5)	407 (3.8; 3.5 to 4.2)	308	NA

CBNAAT = cartridge-based nucleic acid amplification test, CI = confidence interval, NA = not available and NNS = number needed to screen to diagnose on person with TB. ^a^ Uncertain if the denominator is the true number of presumptive TB cases identified after screening.

**Table 3 tropicalmed-06-00206-t003:** Activities and outcomes of the active case finding (ACF) conducted by implementing partner agencies.

	Years	Target Population Mapped	Numbers Screened from Population Mapped (%)	TB Tested in Those with Presumptive TB (%) and among Those Screened [%]	TB Diagnostic Tests			TB Diagnosed (%; 95% CI)	NNS	
Partner Agency					Sputum Positive (%)	X-ray Abnormal (%)	CBNAAT Positive (%)			Anti-TB Treatment Initiated (%)
The Union Axshya Project	2013-2015	NA	8,120,015 households (NA)	225,443/541,406 (41.6) [NA]	21,268/225,443 (9.4)	Nil	Nil	21,268 (9.4; 9.3 to 9.6)	NA	20,589 (96.8)
(The Global Fund)	2015-2017	NA	9,003,299 households (NA)	272,836/535,613 (50.9) [NA]	25,493/272,836 (9.3)	Nil	Nil	25,493 (9.3; 9.2 to 9.5)	NA	24,524 (96.2)
	2018-2020	NA	25,575,009 (NA)	216,075/292,557 (73.9) [0.9]	15,550/216,075 (7.2)	4190/10,136 (41.3)	784/2166 (36.0)	21,012 (9.7; 9.6 to 9.9)	1217	18,373 (87.4)
ICMR TIE-TB (The Global Fund)	2015-2017	6,117,597	55,707 (0.91)	49,998/49,998 (100) [89.7]	2091/49,998 (4.2)	5,272/45,840 (11.5)	NA	4286 (8.5; 8.3–8.8)	13	4286 (100)
KHPT Project THALI (USAID)	2017-2019	NA	NA	21,171/28,473 (74.3) [NA]	1578/NA	NA	30/NA	2247 (10.6; 10.2 to 11.0)	NA	2174 (96.8)
World Health Partners (USAID)	2017–2019	1,707,990	381,761 (22.3%)	6254/6254 (100) [1.6]	451/6254 (7.2)	Nil	Nil	451 (7.2; 6.6 to 7.9)	847	451 (100)
	2018-2020	NA	18,705 (NA)	1155/1398 (82.6) [6.1]	46/279 (16.5)	156/1155 (13.5)	13/192 (6.8)	215 (18.6; 16.5 to 21.0)	87	215 (100)
	2019	NA	20,863 (NA)	501/501 (100) [2.4]	34/501 (6.8)	Nil	Nil	34 (6.8; 4.9 to 9.3)	614	34 (100)
	2018-2019	NA	1389 (NA)	19/42 (45.2) [1.3]	1/19 (5.3)	Nil	Nil	1 (5.3; 0.9 to 24.6)	1389	1 (100)
World Vision (The Global Fund)	2015-2017	3,535,072	1.8 million households (NA)	71,980/NA (NA) [NA]	NA	NA/71,980	NA	34,761 (48.4; 48.0 to 48.7)	NA	34,761 (100)

CBNAAT = cartridge-based nucleic acid amplification test, CI = confidence interval, ICMR = Indian Council for Medical Research, KHPT = Karnataka Health Promotion Trust, NA = not available and USAID = United States Agency for International Development.

**Table 4 tropicalmed-06-00206-t004:** Summary data from the National Tuberculosis Elimination Programme for the active case finding (ACF) activities in 2020 from the states and union territories (ranked by population size) *.

	State/Union Territory (Estimated Population in Millions)	Vulnerable Target Population Mapped from State Population (%)	Numbers Screened from Mapped Target Population (%)	Numbers with Presumptive TB Tested from Those Screened (%)	TB Diagnosed in Those Tested (%; 95% CI)	Number Needed to Screen (NNS)
1	Uttar Pradesh (223.43)	44,019,832 (18.9)	43,255,104 (98.3)	156,980 (0.4)	10,121 (6.5; 6.3 to 6.6)	4274
2	Maharashtra (125.74)	85,791,971 (68.2)	333,161 (0.4)	311,650 (93.5)	12,823 (4.1; 4.0 to 4.2)	26
3	Bihar (124.76)	884,094 (0.7)	13,776 (1.6)	49 (0.4)	7 (1.3; 7.1 to 26.7)	1968
4	West Bengal (99.91)	13,608,540 (13.6)	11,997,372 (88.2)	232,599 (1.9)	1810 (0.8; 0.7 to 0.8)	6628
5	Madhya Pradesh (84.36)	14,668,164 (17.4)	1,070,951 (7.3)	44,341 (4.1)	4912 (11.1; 10.8 to 11.4)	218
6	Tamil Nadu (81.4)	1,148,451 (1.4)	281,122 (24.5)	14,744 (5.2)	395 (2.7; 2.4 to 3.0)	711
7	Rajasthan (79.92)	8,090,518 (10.1)	6,906,255 (85.4)	43,083 (0.6)	1067 (2.5; 2.3 to 2.6)	6473
8	Gujarat (69.76)	65,882,010 (94.4)	50,847,334 (77.2)	121,466 (0.2)	4565 (3.8; 3.7 to 3.9)	11,138
9	Karnataka (68.51)	15,507,273 (22.6)	92,436 (0.6%)	87,505 (94.7)	2939 (3.4; 3.3 to 3.5)	31
10	Andhra Pradesh (52.54)	1,335,818 (2.5)	1,151,885 (86.2)	51,982 (4.5)	1685 (3.2; 3.1 to 3.4)	683
11	Odisha (46.32)	45,292,673 (97.8)	41,965,511 (92.7)	222,198 (0.5)	5116 (2.3; 2.2 to 2.4)	8202
12	Jharkhand (39.48)	14,854,650 (37.6)	15,230 (0.1)	10,731 (70.5)	1891 (17.6; 16.9 to 18.4)	8
13	Telangana (37.92)	754,912 (2.0)	60,632 (8.0)	4822 (8.0)	1207 (25.0; 23.8 to 26.3)	50
14	Assam (35.05)	79,329 (0.2)	15,243 (19.2)	2029 (13.3)	91 (4.5; 3.7 to 5.5)	167
15	Kerala (35.44)	1,171,034 (3.4)	37,685 (3.2)	29,166 (77.4)	802 (2.8; 2.6 to 2.9)	47
16	Punjab (30.67)	4,856,533 (15.8)	4,317,208 (88.9)	5371 (0.1)	529 (9.9; 9.1 to 10.7)	8161
17	Chhattisgarh (30.03)	571,344 (1.9)	7462 (1.3)	6436 (86.3)	170 (2.6; 2.3 to 3.1)	44
18	Haryana (29.44)	9,889,536 (33.6)	8,282,557 (83.8)	30,539 (0.4)	866 (2.8; 2.7 to 3.0)	9564
UT1	Delhi (19.05)	1716 (0.0)	985 (57.4)	256 (26.0)	30 (11.7; 8.3 to 16.2)	33
UT2	Jammu & Kashmir (14.50)	422,954 (2.9)	141,814 (33.5)	15,254 (10.8)	190 (1.3; 1.1 to 2.4)	746
19	Uttarakhand (11.63)	1,291,237 (11.1)	1,785,11 (13.8)	2953 (1.7)	100 (3.4; 2.8 to 4.1)	1785
20	Himachal Pradesh (7.5)	7,485,901 (99.8)	22,709 (0.3)	15,852 (69.8)	595 (3.8: 3.5 to 4.1)	38
21	Tripura (3.96)	198,624 (5.0)	98,845 (49.8)	9084 (9.2)	109 (1.2; 1.0 to 1.5)	906
22	Meghalaya (3.66)	1,435,077 (39.2)	532,359 (37.1)	1064 (0.2)	28 (2.6; 1.8 to 3.8)	19,012
23	Manipur (3.12)	53,336 (1.7)	32,289 (60.5)	3802 (11.8)	52 (1.4; 1.0 to 1.8)	621
24	Nagaland (2.07)	91,005 (4.4)	23,272 (25.6)	1291 (5.5)	23 (1.8; 1.2 to 2.7)	1011
25	Arunachal Pradesh (1.64)	56,236 (3.4)	48,925 (87.0)	2350 (4.8)	73 (3.1; 2.5 to 3.9)	670
26	Goa (1.54)	NA	NA	NA	NA	NA
UT3	Puducherry (1.50)	16,152 (1.1)	10,886 (67.4)	109 (1.0)	5 (4.6; 2.0 to 10.1)	2177
27	Mizoram (1.26)	1,35,399 (10.7)	59,883 (44.2)	293 (0.5)	8 (2.7; 1.4 to 5.3)	7485
UT4	Chandigarh (1.17)	145,297 (12.4)	6962 (4.8)	703 (10.1)	36 (5.1; 3,7 to 7.0)	193
UT5	Dadra & Nagar Haveli; Daman & Diu (0.80)	NA	NA	NA	NA	NA
28	Sikkim (0.66)	62,853 (9.6)	11,034 (17.6)	149 (1.4)	4 (2.7; 1.1 to 6.7)	2759
UT6	Andaman & Nicobar (0.39)	389,615 (99.0)	44,762 (11.5)	432 (1.0)	21 (4.9; 3.2 to 7.3)	2130
UT7	Ladakh (0.34)	5952 (1.7)	5952 (100)	14 (0.2)	0 (0)	NA
UT8	Lakshadweep (0.07)	70,070 (100)	70,070 (100)	509 (0.7)	3 (0.6; 0.2 to 1.7)	23,356
	India (1,377.54)	340,268,106 (24.7)	171,940,182 (50.5)	1,429,806 (0.8)	52,273 (3.66; 3.63 to 3.69)	Median: 906 (IQR 108 to 6550)

* Modified from Annexure 6 in the India TB report 2021 [22]. CI = confidence Interval, IQR = interquartile range and UT = Union territory.

**Table 5 tropicalmed-06-00206-t005:** Challenges in implementing ACF activities as perceived by implementers.

Category	Challenges	Description
Health system challenges leading to pre-diagnostic drop-outs and poor documentation of ACF referrals, TB notifications, treatment outcomes and impact of ACF	Poor access to health facilities	Failure to get tested at health facilities due to the distance and time taken to travel, difficulties in finding transport at convenient times, loss of wages incurred due to travel times.
	Non-availability of all diagnostic tests at peripheral health institutions	Chest radiography and GeneXpert are often not available at one place, but at different levels of health care provision (secondary and tertiary hospitals). This makes it difficult for people to complete the required tests in a day.
	Difficulties in accessing radiography services at secondary hospitals	ACF patients are not considered a priority compared to emergency referrals; shortages in materials, resources and equipment malfunction also contribute.
	Poor documentation of ACF referrals for diagnostic tests Lack of a separate ACF module in the data management system	Referral slips given by field staff for diagnostic tests are often misplaced by patients or are not entered in diagnostic facilities as an ACF referral. Nikshay, the online data management tool, does not specifically link TB notifications identified by the ACF programme with treatment outcomes.
Healthcare provision challenges leading to poor ACF screening and diagnostic outcomes	Poor TB awareness among general population	Despite time and effort spent on advocacy, communication and social mobilisation, large segments of the vulnerable population are unaware of the importance of the ACF programme and were unwilling to fully comply with ACF requirements.
	Obtaining an exact denominator of the population, and the geographical boundaries of areas to be mapped	Difficulty in accurately estimating the number of people residing in geographical areas that are mapped. Figures from the previous census are not dynamic and do not accurately reflect the actual population numbers, or its composition, at the time of ACF activities. In many areas, the geographical boundaries of the areas mapped are not clearly demarcated and often overlapped with adjacent areas.
	Difficulties due to mountainous terrains and hard-to access areas	Areas in the country with mountainous terrains (as in Leh and Kargil in Ladakh), or other hard-to-reach areas, make it difficult for ACF teams to screen all of the mapped populations.
Challenges faced by patients and families leading to poor compliance with ACF requirements	Pressure to undergo screening and testing	People identified with presumptive TB often do not feel unwell. Requests to visit designated diagnostic centres are perceived as undue pressure from the health workers, particularly if they are busy and if the travel involves long distances and time away from productive work
	Non-availability of all family members during screening visits	Not all family members can be present when health workers made home-visits. Available family members may find it difficult to accurately report symptoms in other family members.
	Non-availability of investigations	Patients are dissatisfied when tests are unavailable when they visit diagnostic facilities, and they have to make multiple visits to complete their tests.
	Out-of-pocket expenditure for diagnostic tests	Diagnostic tests are provided free of cost at government-designated facilities. Testing at private diagnostic facilities is often more convenient, but the expenditure involved is considerably greater.

## Data Availability

The data used for this synthesis were from two sources: (1) from the state TB programs and partners in a specific data format shared and (2) from the publicly available annual reports of the National TB Elimination Program. The annual reports of the program are available on the website of the National TB Elimination Program. The data from the states and partners are available from the reviewers upon request.

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
