# Peer review of "Active Case Finding for Tuberculosis in India: A Syntheses of Activities and Outcomes Reported by the National Tuberculosis Elimination Programme"

_tropicalmed, 2021, doi:10.3390/tropicalmed6040206_

Round 1

Reviewer 1 Report

No comments

Author Response

11/10/2021

Mr. Carel Zhang,

Assistant Editor,

MDPI Branch Office, Wuhan

Tropical Med Editorial Office

Dear Mr Zhang,

            Re: Manuscript ID: tropicalmed-1417545

Thank you for the opportunity to revise our manuscript again. We have revised the manuscript according to reviewers’ comments.

As in the previous revision, we have not used track changes since this affects the formatting and table alignment. However, we shall explain each change made in our response to the reviewer’s comments.

  1. Response to reviewer 1:

There were no comments for us to address. We are grateful for previous comments.

  1. Response to reviewer 2:
  • This paper is much improved in that I can now read it and understand what you did.

We thank you for your patience and diligence and for the helpful comments that we feel will improve the reporting of our review.

  • The results are much clearer. but Line 351 pointing out the strategic gaps should be done in the discussion section not the results.  Group all the 6 strategic points together in the discussion to make them more coherent.

Thank you for this suggestion. We had originally listed the gaps under the heading ‘Summary of main findings’ but acknowledge that this section was too long. We therefore omitted this section in our first revision and mentioned the strategic gaps in the results.

We have now removed any mention of the gaps in the results and have grouped them in the discussion under 4.1. The gaps between the expected indicators and outcomes in India’s ACF programme (Lines 465 to 485).

  • You say “the NNS has not be used in the programme to guide ACF planning, so far, and we are highlighting it” yet you only mention NNS once in the discussion. If you think this is important, which it could be, it needs to come over much more clearly in the discussion and in the abstract.

Thank you for this suggestion. The original draft had many mentions of the NNS in the results and the discussion. Following previous suggestions, we had toned down the discussion about the NNS in the first revision.

However, we now highlight the NNS in the abstract (lines 31 to 37), results (lines 418 to 429), discussion (line 461 to 464, 483 to 485; 552-554; 581 to 588).

  • Line 77 I am not clear if the next sentence is expanding on the number of factors, which it should do.

We have rephrased this sentence (Line 76-80).

  • Box 1 I don’t understand 5%(>2%-3)

The NTEPs uses that in its list of indicators. We have now clarified this for clarity Number with sputum smear-positive test results; the expectation is 5% (minimum >2% to 3%).

  • Line 113 The results from the latter source, largely from non-programme ACF activities conducted in India, will be re- ported separately. I am not clear if this is separately in the paper or in a different paper. If it is in this paper you don’t need to say this.

We have clarified that the results from the second source will be reported in a separate publication (Lines 113 -114).

  • Line 177. You say you gathered data from discussions with managers and partner agencies.  This is surely a qualitative component to your study.  You don’t describe the methodology used – was it just informal discussion, using a semi structured questionnaire, were they told the results would be used for research etc.

We have described our methods for gathering data on challenges in implementation in lines 174 to 180 and 449 to 453 and have addressed your concerns in the limitations (lines 597 to 603).

  • Line 184 It would be clearer to say you had responses from 9 of 28 states, two of eight union territories etc. The states are named in table 1 so don't need to be repeated in the text

We have done that (line 184-186)

  • Table 1 the column "TB diagnosed" needs to be wider as it is hard to read, and the alignment in the last column is inconsistent

            We have addressed this.

  • Table 2 the alignment in the last column is inconsistent

            We have done this.

  • Line 335 I don’t understand 11% per million

We have clarified this: ‘The NTEP expects that 110,000 per million vulnerable population (11%) should be mapped for community-based screening’. (Line 336-337).

  • Discussion introduction. Remove 484 to 487.  You want to say what is novel about your paper. Somethink like This is the first paper in India to review the available data on ACF in TB and to use NNS to assess the effectiveness of the different programmes.

Thank you for the suggestion. We have done this (lines 461 to 464).

  • Line 552 and following. This is a different point from data management systems

We have provided more subheadings throughout the discussion that brings together the potential interventions addressing the 6 strategic gaps

These are:

4.2.1 Improving the mapping of vulnerable populations and increasing the uptake of screening. (Lines 489 -514)

4.2.2. Better use of data management systems (Lines 515 to 557)

4.2.3. Moving beyond screening for TB symptoms (Lines 558 to 571)

4.2.4. Increasing the diagnostic yield with ACF (Lines 572 to 588).

  • In the conclusion you say you have identified 6 broad areas where there are gaps. These are not clearly drawn together in the discussion and so much of the impact is lost

We hope that the revised discussion now brings together the 6 strategic gaps and the potential interventions in a clearer and more impactful way.

            We thank you for the opportunity to improve the paper and for your insightful comments.

Reviewer 2 Report

This paper is much improved in that I can now read it and understand what you did.

The results are much clearer.  but Line 351 pointing out the strategic gaps should be done in the discussion section not the results.  Group all the 6 strategic points together in the discussion to make them more coherent

You say “the NNS has not be used in the programme to guide ACF planning, so far, and we are highlighting it” yet you only mention NNS once in the discussion.  If you think this is important, which it could be, it needs to come over much more clearly in the discussion and in the abstract

Line 77  I am not clear if the next sentence is expanding on the number of factors, which it should do.

Box 1 I don’t understand 5%(>2%-3)

Line 113 The results from the latter source, largely from non-programme ACF activities conducted in India, will be re- ported separately. I am not clear if this is separately in the paper or in a different paper.  If it is in this paper you don’t need to say this.

Line 177.  You say you gathered data from discussions with managers and partner agencies.  This is surely a qualitative component to your study.  You don’t describe the methodology used – was it just informal discussion, using a semi structured questionnaire, were they told the results would be used for research etc

Line 184 It would be clearer to say you had responses from 9 of 28 states, two of eight union territories etc.  The states are named in table 1 so don't need to be repeated in the text

Table 1 the column "TB diagnosed" needs to be wider as it is hard to read, and the alignment in the last column is inconsistent

Table 2 the alignment in the last column is inconsistant

Line 335 I don’t understand 11% per million

Discussion introduction.  Remove 484 to 487.  You want to say what is novel about your paper. Somethink like This is the first paper in India to review the available data on ACF in TB and to use NNS to assess the effectiveness of the different programmes

Line 552 and following.  This is a different point from data management systems

In the conclusion you say you have identified 6 broad areas where there are gaps.  These are not clearly drawn together in the discussion and so much of the impact is lost

Author Response

16/10/2021

Mr. Carel Zhang,

Assistant Editor,

MDPI Branch Office, Wuhan

Tropical Med Editorial Office

Dear Mr Zhang,

            Re: Manuscript ID: tropicalmed-1417545

Thank you for the opportunity to revise our manuscript again. We have revised the manuscript according to reviewers’ comments.

As in the previous revision, we have not used track changes since this affects the formatting and table alignment. However, we shall explain each change made in our response to the reviewer’s comments.

  1. Response to reviewer 1:

There were no comments for us to address. We are grateful for previous comments.

  1. Response to reviewer 2:
  • This paper is much improved in that I can now read it and understand what you did.

We thank you for your patience and diligence and for the helpful comments that we feel will improve the reporting of our review.

  • The results are much clearer. but Line 351 pointing out the strategic gaps should be done in the discussion section not the results.  Group all the 6 strategic points together in the discussion to make them more coherent.

Thank you for this suggestion. We had originally listed the gaps under the heading ‘Summary of main findings’ but acknowledge that this section was too long. We therefore omitted this section in our first revision and mentioned the strategic gaps in the results.

We have now removed any mention of the gaps in the results and have grouped them in the discussion under 4.1. The gaps between the expected indicators and outcomes in India’s ACF programme (Lines 465 to 485).

  • You say “the NNS has not be used in the programme to guide ACF planning, so far, and we are highlighting it” yet you only mention NNS once in the discussion. If you think this is important, which it could be, it needs to come over much more clearly in the discussion and in the abstract.

Thank you for this suggestion. The original draft had many mentions of the NNS in the results and the discussion. Following previous suggestions, we had toned down the discussion about the NNS in the first revision.

However, we now highlight the NNS in the abstract (lines 31 to 37), results (lines 418 to 429), discussion (line 461 to 464, 483 to 485; 552-554; 581 to 588).

  • Line 77 I am not clear if the next sentence is expanding on the number of factors, which it should do.

We have rephrased this sentence (Line 76-80).

  • Box 1 I don’t understand 5%(>2%-3)

The NTEPs uses that in its list of indicators. We have now clarified this for clarity Number with sputum smear-positive test results; the expectation is 5% (minimum >2% to 3%).

  • Line 113 The results from the latter source, largely from non-programme ACF activities conducted in India, will be re- ported separately. I am not clear if this is separately in the paper or in a different paper. If it is in this paper you don’t need to say this.

We have clarified that the results from the second source will be reported in a separate publication (Lines 113 -114).

  • Line 177. You say you gathered data from discussions with managers and partner agencies.  This is surely a qualitative component to your study.  You don’t describe the methodology used – was it just informal discussion, using a semi structured questionnaire, were they told the results would be used for research etc.

We have described our methods for gathering data on challenges in implementation in lines 174 to 180 and 449 to 453 and have addressed your concerns in the limitations (lines 597 to 603).

  • Line 184 It would be clearer to say you had responses from 9 of 28 states, two of eight union territories etc. The states are named in table 1 so don't need to be repeated in the text

We have done that (line 184-186)

  • Table 1 the column "TB diagnosed" needs to be wider as it is hard to read, and the alignment in the last column is inconsistent

            We have addressed this.

  • Table 2 the alignment in the last column is inconsistent

            We have done this.

  • Line 335 I don’t understand 11% per million

We have clarified this: ‘The NTEP expects that 110,000 per million vulnerable population (11%) should be mapped for community-based screening’. (Line 336-337).

  • Discussion introduction. Remove 484 to 487.  You want to say what is novel about your paper. Somethink like This is the first paper in India to review the available data on ACF in TB and to use NNS to assess the effectiveness of the different programmes.

Thank you for the suggestion. We have done this (lines 461 to 464).

  • Line 552 and following. This is a different point from data management systems

We have provided more subheadings throughout the discussion that brings together the potential interventions addressing the 6 strategic gaps

These are:

4.2.1 Improving the mapping of vulnerable populations and increasing the uptake of screening. (Lines 489 -514)

4.2.2. Better use of data management systems (Lines 515 to 557)

4.2.3. Moving beyond screening for TB symptoms (Lines 558 to 571)

4.2.4. Increasing the diagnostic yield with ACF (Lines 572 to 588).

  • In the conclusion you say you have identified 6 broad areas where there are gaps. These are not clearly drawn together in the discussion and so much of the impact is lost

We hope that the revised discussion now brings together the 6 strategic gaps and the potential interventions in a clearer and more impactful way.

            We thank you for the opportunity to improve the paper and for your insightful comments.

Round 2

Reviewer 2 Report

this is much improved and can now be published

This manuscript is a resubmission of an earlier submission. The following is a list of the peer review reports and author responses from that submission.

Round 1

Reviewer 1 Report

General comment: The abstract is much too long and should be shortened considerably. Unfortunately, also the manuscript is much too long and difficult to read as the authors try to provide a complete overwiew of every single item of the review. Also the discussion section is inappropriate as it in fact it is indeed a mixture between a summary of the results and – starting from 4.2. – a real discussion.

The full text of sections 2.5 to 2.8. should be transferred to an Online Supplement. In the main manuscript the text should be condensated on a mimimum because it won´t be readable otherwise due to a flood of informations.

Lines 956, 972 or 998: The authors should avoid the frequent use of indefinite words such as “most”, many” or “majority”. Percentages should be used instead. For example, what does “many” mean concretely? 30 percent, 45 percent? 50 percent?

Reviewer 2 Report

This is a potentially important topic for those interested in ACF.

However I found this paper long, very hard to read and difficult get an overview of the findings.  Some results were most clearly stated in the abstract.  

The writing style is part of what makes it hard to read.  Sentences are often excessively long.  Sentences are commonly four lines long e.g. 55 to 59 and  from line 139 to 146 appears to be one sentence.  Sentences of around 14 words are easier to read.

Use of paragraphs to break up text.  There are long sections of unbroken text which could be split into several paragraphs.  The abstract is written as one paragraph which makes it hard to read.

There is too much detail given, particularly in the results.  It feels as though these are the preliminary results being dumped and the key messages have not yet been refined. For example Lines 323 to 328 give me a long list of states where some activity takes place by partner agencies.  I am unclear how I use this data, as if there are lots of smaller projects happening, does it tell me anything about the care in the whole state? Is this paragraph really a result or is it telling me about the projects the partner agency run?

at line 696 do I need to know the % in the different states - is this not a reflection of the way the partner agency works rather than which state it is in?

at line 699 I have no idea what it mean if out of pocket expenditure is reduced by "78% and 254%"  If it normally costs 1000 rupees is it 4 rupees?  Please put it more clearly.

Use of tables is not well thought through.  Are tables the best way to convey what the authors want to show?  could table 6 be a histogram so I can see the differences rather than having to compare figures myself?

What do the authors want me to take away from table 1?  The column titles are unreadable.  Most states score A and some are NA.  Much of the information is repeated in the text.

Table 2 is poorly laid out. Some numbers are split on to different lines.  There is inconsistent use of comma's -eg 1367 and 1,781 - which makes it harder to read.  The text repeats much of the data in the table rather than pulling out the main points the reader should be aware of.  This is repeated in other tables.

The layout of Table 5 doesn't work.  It is not immediately clear there are 2 separate sets of results and I am not clear what gained by having them side by side.

The second table No 7 in the discussion section is inappropriate in the detail and length.  It is not clear if this is based on the updated WHO operational handbook.  If most or all of this information is in the WHO handbook it means it is well known, so why is it being repeated here?

There is no attempt to briefly pull out what are the main messages.

The discussion is too long, poorly thought through and constructed.  The first 150 lines from line 882 are "summarising the results".  The weaknesses section is too long.  There is  relatively little discussion how these finding compare with other studies. 

There is a long section about how the Nishay could be used differently which is not really appropriate as the paper is not about the Nishay.

The relationship between the number screened, sensitivity and specificity of the test and the prevalence of TB affecting the NNS is well known.  It is presented as a new finding whereas it should be included in the introduction and used as part of the basis knowledge.